# Compliance Surveillance of the Tobacco Control Legislation in a Spanish Region and Characterization of Passive Exposure to Tobacco Smoke and E-Cig in Children in Outdoor Spaces

**DOI:** 10.3390/healthcare10040717

**Published:** 2022-04-13

**Authors:** Laura Jovell, Ana Díez-Izquierdo, Juan Carlos Martín-Sánchez, Àurea Cartanyà-Hueso, Adrián González-Marrón, Cristina Lidón-Moyano, Jose M. Martínez-Sánchez

**Affiliations:** 1Group of Evaluation of Health Determinants and Health Policies, Faculty of Medicine and Health Science, Universitat Internacional de Catalunya (UIC-Barcelona), 08195 Sant Cugat del Valles, Spain; ljovell@uic.es (L.J.); jcmartin@uic.es (J.C.M.-S.); acartanya@uic.es (À.C.-H.); agonzalezm@uic.es (A.G.-M.); clidon@uic.es (C.L.-M.); jmmartinez@uic.es (J.M.M.-S.); 2Pediatric Allergy and Pulmonology Section, Department of Pediatrics, Hospital Universitari Vall d’Hebron, 08035 Barcelona, Spain

**Keywords:** second hand smoke, children, tobacco control legislation, smoking signage, SHS

## Abstract

(1) Background: Exposure to environmental tobacco smoke has decreased in recent years in Spain, due to the implementation of tobacco control policies. However, there is no regulation that protects against second-hand smoke (SHS) in outdoor environments. Our goal is to describe the smoking prohibition signage in public spaces and to characterize tobacco consumption in outdoor environments describing the SHS exposure in children. (2) Methods: A cross-sectional study using direct observation was carried out with a convenience sample (n = 179) that included hospitality venues with terraces, schools and healthcare facilities in the municipality of Sant Cugat del Vallès (Barcelona, Spain). The observations were made without notifying the owners by one single field researcher between April and June 2018. The variables were evaluated by signage and signs of tobacco consumption (ashtrays, cigarette butts and presence of smokers). (3) Results: Smoke-free zone signage outside public spaces was present in 30.7% of all venues, with only 50.9% correctness. When analysing terraces of hospitality venues, in 35.8% of them there were children present with 66.7% of tobacco consumption. (4) Conclusions: Our results show a low prevalence of antismoking signage, without an impact on tobacco consumption regardless of the presence of children.

## 1. Introduction

Second-hand smoke (SHS) exposure is known to be a cause of lung cancer and coronary heart disease. Its status as an important public health hazard still prevails as one of the top causes of mortality worldwide, without a risk-free level of SHS exposure. However, SHS causes premature death and disease not only in adults but in children as well, and increases the risk of acute respiratory tract infection (RTI), sudden infant death syndrome (SIDS) and more severe asthma, among other harmful effects [1]. Despite new and improved measures in tobacco control, implemented in recent years, millions are still exposed to SHS [1]. Tobacco control has shown health benefits in the paediatric population, emphasized by a decrease in preterm births and perinatal mortality [2,3], fewer emergency room (ER) visits associated with asthma [4,5], and a reduction in hospital admissions for RTI [6,7,8]. The aim of this study was to describe antismoking signage in public spaces and tobacco consumption outdoors, including SHS exposure in children.

In the last decades, there has been an increase in the implementation of smoke-free laws in indoor public places and workplaces in many countries [9]. The MPOWER package released by the WHO [10] assists in implementing the WHO-FCTC treaty [11]. One of its measures is monitoring tobacco use and prevention policies, including the impact of policy interventions such as smoke-free zone signage [9].

Following article eight of the WHO-FCTC, major steps were taken in Spain against the tobacco epidemic with the implementation of Law 42/2010 [12]. It bans smoking in hospitality venues and other outdoor areas (e.g., healthcare campus premises, primary schools and secondary school facilities) but does not provide a proper description or picture of the correct smoke-free signs. This law complements previous legislation [13] that declared as smoke-free areas all public and work places as well, in view of the association of SHS exposure with harmful effects [1,14].

Despite later improvements, new challenges still exist in Spain regarding tobacco control in open spaces [15], which are not included in the current legislation [12,16]. The need to properly define the concept of premises and direct access points into healthcare facilities and schools as smoke-free zones still lingers. Moreover, the definition of terraces in bars and restaurants remains doubtful as well, regarding what is, and what is not, an outdoor space [12]. It has been described that tobacco smoke from immediate entrances drifts indoors despite the smoking prohibition inside, exposing people to SHS [17]. Additionally, some municipal jurisdictions have limited tobacco use in certain places where minors are present [15].

SHS is not limited to exposure to conventional tobacco smoke but also to the aerosols of e-cigarettes. The prevalence of their use in Spain is high, especially in restaurants and bars [18]. However, although included in the legislation, their use is not yet fully regulated in outdoor areas [19]. For instance, in December 2018, the US Surgeon General declared e-cigarette use among youth an epidemic in the United States [20].

The objective of this study was to describe the smoking prohibition signage in public spaces (healthcare facilities, hospitality venues, schools) and to characterize tobacco and e-cig consumption in outdoor environments (terraces of bars/restaurants and educational centres) describing the SHS exposure in children.

## 2. Materials and Methods

### 2.1. Study Design and Data Collection

This was a descriptive cross-sectional study carried out in the municipality of Sant Cugat del Valles (Barcelona, Spain), which included hospitality venues and public places, using direct observation (n = 179). The fieldwork occurred between April and June of 2018. Data were collected on weekdays outside school hours (to ensure the presence of children) and on weekends between 9 A.M. and 9 P.M. each day. The monitoring took place without notifying or warning the owners or other parties involved to limit observer bias (Hawthorne effect).

It included 58.9% of all hospitality venues with terraces, and almost all schools (24 out of 26) and healthcare facilities (10 out of 11) in the area. All of the observations were made by a single field researcher to ensure uniformity in the observations, given that there is inter-observer agreement in direct observation studies, which are a good resource for monitoring smoking [21]. The observer completed the tobacco worksheet walking along the area and, upon arriving at a location, stood next to the door recording the different variables, spending around 15 min at each venue.

Except for hospitality venues, we extracted the list of locations from the city council webpage [22]. Among the eligible hospitality venues spread around the municipality, we selected those areas with a higher density of establishments. This criterion was implemented to improve the likelihood of finding open venues since there is a significant amount of ground surface in the area that is either forest or residential area.

The inclusion criterion was the presence of at least one person in the terrace for hospitality venues. The number of healthcare centres, schools, public administration offices and transport facilities was small; therefore, they were all included. For each location, we registered at the door that served as the main entrance and the adjacent hall. In schools, the door recorded was the one used by the students to access the building. Additionally, in healthcare facilities, different doors were considered (both main entrance and emergency entrance) since the Catalan Network of Smoke-free Hospitals supports the application of Law 42/2010 to implement smoke-free campuses [23]. This regional network only includes those hospitals and health centres that voluntarily participate in this smoke-free initiative.

The approval from an ethics committee was not required because this was an observational study that did not collect the participants’ private information; moreover, there were no manipulations or invasive measurements.

### 2.2. Study Variables

A data collection sheet was created to include the different variables of the study (Figure A1—Appendix A). We defined the different variables about signage and signs of tobacco consumption (i.e., ashtrays, cigarette butts, and presence of smokers).

The types of locations included hospitality venues (including bars, restaurants, cafeterias, and bakeries with terraces), healthcare facilities, primary schools, and secondary schools. Regarding the physical characteristics of hospitality venues, we indicated whether the establishment had a terrace or not and the presence of sidewalls (yes/no). We considered sidewalls as any permanent or temporary structure that impeded lateral airflow, regardless of their full attachment to the roof.

The following variables were recorded both outdoors (entrance and terrace if present) and indoors (hall) at the different locations, except in schools where the inside was inaccessible. According to the local, national, and European guidelines, we documented the presence of smoke-free zone signage as well as whether it was correct or not, both for conventional tobacco and e-cigarettes [24,25]. We documented the presence of cigarette butts (yes/no), the presence of ashtrays (yes/no), and the presence of smokers (yes/no). If smokers were present, we registered whether they were using conventional/manufactured, roll-your-own cigarettes or e-cigarettes. In addition, we recorded whether there were children (yes/no) present at the same time as smokers.

### 2.3. Statistical Analyses

For the data analysis, we included all locations, and later stratified them by type of setting (i.e., hospitality venues, schools, and healthcare facilities). To describe the variables, we used absolute frequencies and percentages, such as signage presence on the doors of public establishments and its correctness; as well as the percentage of tobacco consumption in terraces of hospitality venues and at the entrance of educational centres according to the presence of children (up to 5 years old) stratified by covariates. All data of hospitality venues were stratified by age of the child, conventional tobacco door signage and its correctness, cigarette butts, ashtrays, type of terrace, time of day, and day of the week.

The percentages of independent groups were compared using the Chi-Squared test, and the Fisher’s exact test. *p*-values below 0.05 were considered statistically significant. We performed all analyses with R statistical software v. 3.1.0.

## 3. Results

Regarding the signage on the doors of public establishments (Table 1), 30.7% of all venues displayed exterior smoke-free zone signage for conventional tobacco, while only 50.9% of it was correct according to the local, national, and international guidelines [24,25]. The healthcare facilities showed the highest percentage of smoke-free zone signage (80.0%, n = 10), although only 37.5% was correct. In contrast, we observed the lowest signage percentage in publicly administered facilities (city halls, police stations, libraries, theatres, and municipal offices), which was non-existent. We found a low percentage of smoke-free zone signage in schools, with no significant difference between primary schools and secondary schools (20.0% and 17.7% respectively), with a correctness percentage of 40.0% and 33.3%, correspondingly.

On the subject of e-cigarettes, the number of all venues that had exterior smoke-free zone signage was small (n = 3, 1.7%). However, in those venues, the e-cigarette signage displayed 66.7% correctness. The highest percentage of signage was found in healthcare facilities, and the lowest percentage (0.0%) was observed in schools, publicly administered facilities, and transport facilities (Table 1).

The prevalence of tobacco consumption in the terraces of hospitality venues and at the entrances of educational centres (Table 2) was analysed according to the presence of children (up to 5 years old) and stratified by covariates. There were smokers in 66.97% of all the terraces in hospitality venues, of which 28.4% had conventional tobacco door signage and only 48.4% was correct (data not shown (DNS)). Terraces with ashtrays presented 80.3% of tobacco consumption compared to 36.4% that did not (*p* < 0.001). In those hospitality venues having correct signage, the percentage of tobacco consumption was 80.0%, which decreased to 66.7% when there were children present in the terraces. The percentages of tobacco consumption in terraces were similar whether there were children present or not for the rest of covariates. Regarding the presence of ashtrays, we found statistically significant differences in the percentage of tobacco consumption, being higher in those terraces where there were ashtrays. This pattern was kept even after stratifying by the presence of children in terraces (Table 2). In the terraces of hospitality venues where children were present (n = 45), 31.1% of smokers were at the same table as the children (DNS).

At the entrance of educational centres, we observed a prevalence of 16.7% of tobacco consumption (n = 24). Among those schools with exterior signage, its correctness showed no tobacco consumption compared to 33.3% when the signage was not correct (Table 2). In 95.8% of schools (DNS), cigarette butts were present at the entrance, and 17.4% of those schools had tobacco consumption at the entrance.

## 4. Discussion

### 4.1. Main Finding of This Study

Our results revealed a low prevalence of antismoking signage overall, as well as against e-cigarette consumption, in Catalonia. Two thirds of tobacco consumption regardless of children’s presence was registered in hospitality venues, comprising a low percentage of antismoking signage in schools with 16.7% tobacco consumption, a banned practice there.

Antismoking signage and its correctness did not sway people’s smoking habits. Cigarette butts were present in all but one establishment, a non-negligible number [1]. The presence of ashtrays showed a higher prevalence of smokers in their terraces when available, with statistically significant differences. Smokers may be influenced by ashtrays, since they smoked more in terraces with than without ashtrays, as previously suggested [26]. This might indicate hospitality venues’ owners could impact customers’ behaviour regarding their smoking habits in terraces by providing ashtrays or not.

Smokers were registered in terraces with and without children alike, despite social awareness of SHS exposure’s harmful effects [14,27,28], implying people might not consider children when smoking. Terraces are not properly regulated by law because they are considered outdoor spaces. However, there are limitations in the definition of “outdoor”, as sidewalls are taken into account, since those spaces can acquire indoor qualities, such as not letting the smoke out. Results suggest smokers and hospitality venues do not consider whether existing sidewalls allow smoking, besides the presence of children in those terraces.

Schools had smokers; although few, they should be non-existent [12]. People smoked regardless of the signage, which might be due to a lack of awareness of the direct harms from SHS [29]. However, there seems to be a correlation with the school enforcing the law, since when the signage was correct, there were no smokers. This seems to reinforce that correct legislation and due implementation would help reduce smokers around schools. In other countries smoking is banned within a specified distance from entrances and buildings, such as in most jurisdictions in Australia and Canada [30].

In general, seven out of ten places did not display any prohibition signage, thus not reminding people that smoking was banned on their premises. Correct signage was displayed in only half of the studied places. Spanish law [12] states that smoking is not allowed indoors, including hospitality venues, and other outdoor areas such as healthcare campuses premises and school facilities. Although the law shows consideration for the presence of antismoking signage, it does not specify the proper way to indicate it, or the correct pictures. Aside from the international no smoking sign widely used in most countries (Figure A2—Appendix B), the national and local webpages provide guidelines indicating the correct and accepted antismoking signs in the region [24,25]. Recently, the government has been contemplating a legislative amendment depending on results from the new inspections campaign about compliance with the antismoking ban in 2020 [31].

Healthcare facilities in some Spanish regions, such as Catalonia [23], have their own antismoking signage. This explains the high prevalence of signage in healthcare facilities (80%), despite its lack of correctness, since these places are usually frequented by health-sensitive individuals. Spain should follow the initiative of other countries such as Norway, Iraq, Israel or Ontario in Canada and extend the law to include hospital grounds and surrounding areas [30].

Schools display a very low percentage of prohibition signage (20%) despite being required by law, which includes the outside of all educational centres. Despite children being protected by the law, only two out of five signs were correct, meaning schools are not following the regulations regarding SHS exposure prevention.

### 4.2. What Is Already Known on This Topic

E-cigarette use has been on the rise over the last few years in many countries, including the US, UK, France, Germany, and Spain [20,32]. Their consumption is not entirely regulated like conventional tobacco in other countries, although the latest reports indicate their aerosols are equally as harmful [18]. Therefore, e-cigarette use should be fully legislated to protect people from exposure to their SHS. The signage banning e-cigarettes was very minimal (1.7%), including only a couple of healthcare facilities and one restaurant, which is hardly representative, as the proper signage is included in the legislation. This lack of compliance could be related to a lack of awareness about the harmful effects of e-cigarette SHS exposure [33]. However, non-smokers’ exposure is not limited to SHS but also to the harmful health consequences from residual tobacco smoke pollutants (defined as third-hand smoke (THS)) that remain on surfaces after tobacco has been smoked [34,35].

### 4.3. What This Study Adds

One third of terraces with children present had smokers seated at the same table as the children, showing a disregard among caregivers for their own children’s health. Furthermore, children do not have control over their environments or an awareness of the dangers, meaning they need protection by a third party, their caregivers, or legislation [4]. These kinds of legislation should be implemented in Spain to protect vulnerable populations (i.e., children, pregnant women, the elderly), reinforcing the idea of promoting fully smoke-free zones around schools and healthcare environments. The smoking prohibition should apply equally to e-cigarettes in all smoke-free zones as well [18]. Awareness campaigns would also help educate the public about the relevance of this issue.

Since the implementation of the smoking ban [13], there has been considerable progress with the regulation of smoking in Spain [12,16]. However, SHS exposure remains in public [36] and private settings [37], carrying harmful health consequences [1,38], especially for children who should be protected by law. Therefore, new and updated legislation is necessary to create correct and standardized signage and avoid SHS exposure, thus protecting schools and hospital surroundings, as well as limiting tobacco use in places where minors are present. We believe broader smoke-free legislation will be the basis for future progress similar to that made in other countries.

### 4.4. Limitations

One limitation of this study was that routes were not fully randomized due to logistic field-work reasons. Since Sant Cugat del Vallès has a large zone of forested and residential areas, the areas selected had high population density. Moreover, the age variable could have classification bias, which is why it was collected using intervals. However, this study also contained strengths such as the inclusion of a high percentage of municipal locations (hospitals, terraces, schools). Furthermore, to avoid apprehension bias, monitoring was conducted without notifying the owners of venues or other interested parties.

## 5. Conclusions

The existing low prevalence of antismoking signage did not impact people’s behaviour in relation to tobacco consumption, regardless of the presence of children, thus making it urgent to strengthen the antismoking signage. It is necessary to better educate the public about the effects of SHS and THS exposure, especially in children. In fact, the current antismoking law should be revised regarding an outdoor smoking ban to protect vulnerable populations.

## Figures and Tables

**Table 1 healthcare-10-00717-t001:** Percentages of exterior smoke-free zone signage and correctness on the doors of public establishments for conventional tobacco and electronic cigarette consumption.

		Conventional Tobacco			Electronic Cigarettes		
	n	Signage	n	Correctness *	Signage	n	Correctness
Total	179	30.7%	55	50.9%	1.7%	3	66.7%
Primary education	25	20.0%	5	40.0%	0.0%	0	0.0%
Secondary education	17	17.7%	3	33.3%	0.0%	0	0.0%
Hospitality venues	111	28.8%	32	46.9%	0.9%	1	0.0%
Healthcare facilities	10	80.0%	8	37.5%	20.0%	2	100.0%
Public administration	12	0.0%	0	-	0.0%	0	-
Transport	9	55.6%	5	100.0%	0.0%	0	-

* Only when the signage is present.

**Table 2 healthcare-10-00717-t002:** Percentage of tobacco consumption in hospitality venues’ terraces and at the entrances of educational centres according to the presence of children (up to 5 years old) by covariates.

		Hospitality Venues’ Terraces							Entrances of Educational Centres		
		n	All Terraces	*p*-Value	Terraces with Children (n = 39)	*p*-Value	Terraces without Children (n = 70)	*p*-Value	n	Total	*p*-Value
Total		109	66.97%	-	66.67%	-	67.14%	-	24	16.67%	-
Age of the child				-		1.000 ^1^		-			1.000 ^2^
	<1 year old	15	-		66.67%		-		4	0.00%	
	1–5 years old	24	-		66.67%		-		20	20.00%	
Conventional tobacco door signage				0.433 ^1^		1.000 ^1^		0.354 ^2^			1.000 ^2^
	Yes	31	74.19%		68.75%		80.00%		5	20.00%	
	No	78	64.10%		65.22%		63.64%		19	15.79%	
Correct signage *				0.685 ^2^		1.000 ^2^		0.525 ^2^			1.000 ^2^
	Yes	15	80.00%		66.67%		88.89%		2	0.00%	
	No	16	68.75%		70.00%		66.67%		3	33.33%	
Cigarette butts				1.000 ^2^		1.000 ^2^		1.000 ^2^			1.000 ^2^
	Yes	107	67.29%		66.67%		67.65%		23	17.39%	
	No	2	50.00%		-		50.00%		1	0.00%	
Ashtrays				<0.001 ^1^		0.008 ^2^		0.002 ^1^			1.000 ^2^
	Yes	76	80.26%		81.48%		79.59%		0	-	
	No	33	36.36%		33.33%		38.10%		24	16.67%	
Type of terrace				0.370 ^1^		1.000 ^2^		0.313 ^2^			-
	With sidewalls	22	77.27%		70.00%		83.33%			-	
	Without sidewalls	87	64.37%		65.52%		63.79%			-	
Time of day				0.531 ^1^		0.589 ^2^		0.733 ^1^			1.000 ^2^
	Morning	22	59.09%		50.00%		61.11%		20	20.00%	
	Afternoon	87	68.97%		68.57%		69.23%		4	0.00%	
Day of the week				0.339 ^1^		0.157 ^1^		0.982 ^1^			-
	Weekdays	51	72.55%		76.00%		69.23%			-	
	Weekends	58	62.07%		50.00%		65.91%			-	

^1^ Chi-squared test. * Only when the signage is present. ^2^ Fisher test.

## Data Availability

Not application.

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
