# Peer review of "Compliance Surveillance of the Tobacco Control Legislation in a Spanish Region and Characterization of Passive Exposure to Tobacco Smoke and E-Cig in Children in Outdoor Spaces"

_healthcare, 2022, doi:10.3390/healthcare10040717_

Round 1

Reviewer 1 Report

  1. There is only one researcher in a place to observe. How to ensure the accuracy of the observation results?
  2. How to accurately observe children < 5 years old through the naked eye?
  3. The method section mentioned recording the distance between smokers and children, but the results section did not show this variable.
  4. The research of the article is not deep enough, and the content of the result part is relatively thin.

Author Response

Reviewer: 1

General comments.

There is only one researcher in a place to observe. How to ensure the accuracy of the observation results? How to accurately observe children < 5 years old through the naked eye?

Thank you very much for the comment. This it is a cross-sectional study, so there is no contact or follow-up of the participants. Besides, the main objective of the study is to describe the smoking prohibition signage in public spaces (healthcare facilities, hospitality venues, schools) and to characterize tobacco consumption in outdoor environments (terraces of bars/restaurants and educational centres), describing the second-hand smoke exposure in children overall. Therefore, to ensure uniformity and based on the literature cited below, observation in children was done through the naked eye. It is easy to differentiate between a 3-5 years old child and a 6-7 years old one. Nevertheless, it is possible that unfortunately in some cases we have erroneously misclassified the age limit. However, it would have been done equally in both groups and we do not believe it will have any repercussions, as the possible derivative bias will be the same in both age groups and therefore should not change or impact the results. In addition, the priority was to not communicate or interfere with the observed subjects to avoid any apprehension bias or the Hawthorne effect.

Moreover, as the reviewer suggested, the manuscript has been modified and we have clarified it proving more information about the study in the methods section as follows:

“This was a descriptive cross-sectional study carried out in the municipality of Sant Cugat del Vallés (Barcelona, Spain) which included hospitality venues and public places, using direct observation (n=179). The fieldwork occurred between April and June of 2018. Data were collected on weekdays outside school hours (to ensure the presence of children) and on weekends between 9 a.m. and 9 p.m each day. The monitoring took place without notifying or warning the owners or other parties involved to limit observer bias (Hawthorne effect).

It included 58.9% of all hospitality venues with terrace, and almost all schools (24 out of 26) and healthcare facilities (10 out of 11) in the area. All the observations were made by one single field researcher to ensure uniformity in the observations, given that there is inter-observer agreement in direct observation studies, which are a good resource for monitoring smoking [21]. The observer completed the tobacco worksheet walking along the area and, upon arriving at a location, stood next to the door recording the different variables, spending around fifteen minutes at each venue.”

_______________

References updated in the manuscript:

  1. Martínez-Sánchez JM, Curto A, Fernández E. Agreement between two observers in the measurement of smoking and use of safety belt and cell phones in vehicles. Gaceta Sanitaria. 2012; 26 (1): 91–93.

The method section mentioned recording the distance between smokers and children, but the results section did not show this variable.

We agree with the statement. Sometimes, there are studies that record many more variables than then are used when analysing the data to answer the questions of the study, the variable distance was collected in our study but then it was not incorporated into the analysis because we have to select the variables more interesting for us. Therefore, as was suggested by the reviewer, the variable distance between smokers and children has been removed from the text, and we have modified the methods section as follows:

“The following variables were recorded both outdoors (entrance and terrace if present) and indoors (hall) of the different locations, except in schools where the inside was inaccessible. According to the local, national, and European guidelines, we documented the presence of smoke-free zone signage as well as whether it was correct or not, both for conventional tobacco and e-cigarettes [24,25]. We documented the presence of cigarette butts (yes/no), the presence of ashtrays (yes/no) and the presence of smokers (yes/no). If users were present, we registered if they were conventional/manufactured, roll-your-own cigarettes or e-cigarettes. Besides, we recorded whether there were children (yes/no) present at the same time as smokers.”

The research of the article is not deep enough, and the content of the result part is relatively thin.

Thank you very much for the suggestions. Ours is a cross-sectional study that concludes that there is a low prevalence in compliance with anti-tobacco signage despite current legislation, highlighting the importance in the paediatric population and its relevance to public health. In addition, this study shows present-day data evaluating the signage that exists at this time and reflecting the behaviour of smokers and establishments, including both public and private, which should guarantee that the current legislation is complied with.

Moreover, as aforementioned and as the reviewer suggested, we have clarified the text in the discussion section as follows:

Our results reveal a low prevalence of antismoking signage overall, as well as against e-cigarette consumption, in Catalonia. Two thirds of tobacco consumption regardless of children’s presence was registered in hospitality venues, comprising a low percentage of antismoking signage in schools with 16.7% tobacco consumption, a banned practice there.

Antismoking signage and its correctness did not sway people smoking habits. Cigarette butts were present in all but one establishment, a non-negligible number [1]. The presence of ashtrays showed a higher prevalence of smokers in their terraces when available, with statistically significant differences. Smokers may be influenced by ash-trays, since they smoked more than in terraces with no ashtrays, as previously suggested [26]. This might indicate hospitality venues’ owners could impact customers’ behaviour regarding their smoking habits in terraces by providing or not ashtrays.”

Reviewer 2 Report

Thanks for inviting me to review this manuscript. This study mainly investigates the smoking prohibition signage in public spaces and tobacco consumption in outdoor environments of Sant Cugat del Vallès, Barcelona, Spain. The authors concluded that the low coverage of antismoking signage in this city might not enough to reduce passive exposure to tobacco of children in outdoor spaces. I think the topic is meaningful and the manuscript was well written and I only have minor concerns for the authors to consider.

  1. Line 101, I think the reason why this study did not require an ethics committee is due to that they did not collect the participants’ private information, but not because of this was an observational study.
  2. There should be a section in the methods to describe that how they recruit participants, how they ask the question (questionnaire or just oral questioning of a respondent), and what specific questions in the list.
  3. I would suggest the authors to conduct several association analyses, for example, the relations between antismoking signage and the ashtrays, cigarette butts and presence of smokers, etc.

Author Response

Reviewer: 2

Comments to the Author

Thanks for inviting me to review this manuscript. This study mainly investigates the smoking prohibition signage in public spaces and tobacco consumption in outdoor environments of Sant Cugat del Vallès, Barcelona, Spain. The authors concluded that the low coverage of antismoking signage in this city might not enough to reduce passive exposure to tobacco of children in outdoor spaces. I think the topic is meaningful and the manuscript was well written and I only have minor concerns for the authors to consider.

Thank you very much for the kind comments to our work.

Minor issues

  1. Line 101, I think the reason why this study did not require an ethics committee is due to that they did not collect the participants’ private information, but not because of this was an observational study.

Thank you very much for the kind comments. We have given more details about the ethical aspects and modified it as suggested as follows:

The approval from an ethics committee was not required because this was an observational study that did not collect the participants’ private information, besides there being no manipulations or invasive measurements.”

  1. There should be a section in the methods to describe that how they recruit participants, how they ask the question (questionnaire or just oral questioning of a respondent), and what specific questions in the list.

Thank you very much for the kind comments. Though it is a good suggestion, our study is an observational study with a cross-sectional design, in which direct observation was made of individuals passing randomly through the observed site, with no contact with the participants or follow-up over time. Therefore no participants have been recruited and neither written nor oral questionnaires have been distributed, hence this section of methods would not be applicable to our study.

Moreover, we also included as elsewhere suggested one of the completed ‘tobacco worksheets’ (line 83) as an example of the specific items in the list registered during the observation carried out with this method. We have added an Appendix with the Tobacco worksheet for data collection, as follows:

Appendix B

Figure 2. Tobacco worksheet for data collection.

  1. I would suggest the authors to conduct several association analyses, for example, the relations between antismoking signage and the ashtrays, cigarette butts and presence of smokers, etc.

Thank you very much for the suggestions. We carried out the analysis that included correlation between the presence of signage and the presence of ashtrays to the presence of children on these terraces. Unfortunately, in our case, the suggested analysis was not performed, so we do not have that information but it is not directly related to the purpose of our study, which is centred on the presence of children.

Moreover, the main objective of the study is to describe the smoking prohibition signage in public spaces (hospitals, hospitality venues, schools) and to characterize tobacco consumption in outdoor environments (terraces of bars/restaurants and educational centres), describing the second-hand smoke exposure in children overall. These findings are described in the methods section, after clarifying some concepts, as follows:

On the subject of e-cigarette, the number for all the venues that had exterior smoke-free zone signage was small (n=3, 1.7%). However, in those venues the e-cigarette signage presented a 66.7% of correctness. The highest percentage of signage was also found in healthcare facilities and the lowest percentage (0.0%) was observed alike in schools, public administration organisms and transport facilities (Table 1).

The prevalence of tobacco consumption in the terraces of hospitality venues and at the entrance of educational centres (Table 2) was analysed according to the presence of children (up to five years old) and stratified by covariates. There were smokers in 66.97% of all the terraces in hospitality venues, of which 28.4% have conventional tobacco door signage and only a 48.4% is correct (data not shown [DNS]). Terraces with ashtrays presented an 80.3% of tobacco consumption compared to 36.4% that did not (p<0.001). In those hospitality venues having correct signage, the percentage of tobacco consumption is 80.0%, which decreases to 66.7% when there were children present in the terraces. The percentages of tobacco consumption in terraces were similar whether there were children or not for the rest of covariates. Regarding the presence of ashtrays, we found statistically significant differences in the percentage of tobacco consumption, being higher in those terraces where there were ashtrays. This pattern was kept even after stratifying by the presence of children in terraces (Table 2). In the terraces of hospitality venues where children were present (n=45), 31.1% of smokers were in the same table as the children (DNS).

At the entrance of educational centres, we observed a prevalence of 16.7% of tobacco consumption (n=24). Among those schools with exterior signage, its correctness showed none tobacco consumption compared to a 33.3% when the signage was not correct (Table 2). In 95.8% of schools (DNS), cigarette butts were present at the entrance, and 17.4% of those schools had tobacco consumption at the entrance.”

Moreover, this concept is explored further in the discussion of the article as follows:

“Antismoking signage and its correctness did not sway people smoking habits. Cigarette butts were present in all but one establishment, a non-negligible number [1]. The presence of ashtrays showed a higher prevalence of smokers in their terraces when available, with statistically significant differences. Smokers may be influenced by ash-trays, since they smoked more than in terraces with no ashtrays, as previously suggested [26]. This might indicate hospitality venues’ owners could impact customers’ behaviour regarding their smoking habits in terraces by providing or not ashtrays.”

Reviewer 3 Report

Dear authors,

This is a very important topic, and you conducted a well-designed research. 

Introduction and materials and methods are clear

Results: Table 2 is not very clear. Can you also give 

"There were smokers in 67.0% of all the 156 terraces in hospitality venues, of which 28.4% have conventional tobacco door signage 157 and only a 48.4% is correct (data not shown [DNS])." in the table..

International regulations and comparison to the other countries are more emphasized in discussion

Author Response

Reviewer: 3

Comments to the Author

Dear authors, This is a very important topic, and you conducted a well-designed research. Introduction and materials and methods are clear.

Thank you very much for the kind comments.

General comments.

Results: Table 2 is not very clear. Can you also give "There were smokers in 67.0% of all the 156 terraces in hospitality venues, of which 28.4% have conventional tobacco door signage 157 and only a 48.4% is correct (data not shown [DNS])." in the table…

International regulations and comparison to the other countries are more emphasized in discussion.

Thank you very much for the comment, and giving us the opportunity to clear up the confusion. This paragraph includes information calculated based on data that appears in the same table 2, but since it does not compare terraces with and without children it is stated as data not shown. The value of smokers in all the terraces in hospitality venues of 67.0% does appear rounded in Table 2 as 66.97%. Conventional tobacco door signage, considered as the international sign for conventional tobacco according to international guidelines, was present in a 28.4% of all terraces, a value calculated by dividing the number of hospitality venues that had conventional tobacco door signage (n=31) by the total number of terraces (n=109). Finally the percentage of terraces with the correct conventional tobacco door signage was only 48.4%, which was calculated by dividing the number of hospitality venues that had correct signage (n=15) by the total number of hospitality venues that had conventional tobacco door signage (n=31).

Therefore, given the aforementioned calculations and specifications, as the reviewer suggested, we have clarified the text in the results section as follows:

“The prevalence of tobacco consumption in the terraces of hospitality venues and at the entrance of educational centres (Table 2) was analysed according to the presence of children (up to five years old) and stratified by covariates. There were smokers in 66.97% of all the terraces in hospitality venues, of which 28.4% have conventional tobacco door signage and only a 48.4% is correct (data not shown [DNS]). Terraces with ashtrays presented an 80.3% of tobacco consumption compared to 36.4% that did not (p<0.001). In those hospitality venues having correct signage, the percentage of tobacco consumption is 80.0%, which decreases to 66.7% when there were children present in the terraces. The percentages of tobacco consumption in terraces were similar whether there were children or not for the rest of covariates. Regarding the presence of ashtrays, we found statistically significant differences in the percentage of tobacco consumption, being higher in those terraces where there were ashtrays. This pattern was kept even after stratifying by the presence of children in terraces (Table 2). In the terraces of hospitality venues where children were present (n=45), 31.1% of smokers were in the same table as the children (DNS).”

Moreover, we have clarified the tobacco door signage following the international guidelines, considering the Appendix A is the international sign used to indicate the antismoking ban, and in accord with the current legislation of each country, as follows:

“In general, seven out of ten places did not display any prohibition signage, thus not reminding people smoking is banned in their premises. The signage rightness is only present in half of the studied places. The Spanish Law [12] states that smoking is not al-lowed indoors, including hospitality venues, and other outdoor areas such as healthcare campuses premises and school facilities. Although the consideration of antismoking signage presence, it does not specify the proper way to indicate it, nor the correct pictures. Besides the international no smoking sign widely used in most countries (Figure A1), the national and local webpages provide guidelines indicating the correct and accepted antismoking signs in the region [24,25]. Recently, the Government is contemplating a legislation addendum depending on results from the new inspections campaign about compliance of the antismoking ban during 2020 [31].”

Reviewer 4 Report

This is a very interesting topic. The research on smoking behaviours, smoking control legislation, and its implications for public health, including children, in outdoor spaces is a critical issue in recent environmental and public health studies. The manuscript reports on observational records in Barcelona, Spain.

Recommendations for practice and policy: A potential key contribution of this study could be about offering recommendations for practice and policymaking; for example, calling for informing stakeholders, including public and healthcare managers, through media and education regarding the effects of SHS exposure and/or the need to consider proximity distance limitations to create a fully smoke-free zone around schools and healthcare environments etc.

Suggestion for the authors: It would be helpful to include one of the completed ‘tobacco worksheets’ (line 83) as an example of observation carried out with this method (perhaps in the appendix). This helps the reader understand the method and details of data collection.

Minor issues: consistency in the citation style

Page 1, line 35: 1 should read [1]

Page 1, line 44: 9 should read [9]

Page 2, line 47: 9 should read [9]

Author Response

Reviewer: 4

Comments to the Author

This is a very interesting topic. The research on smoking behaviours, smoking control legislation, and its implications for public health, including children, in outdoor spaces is a critical issue in recent environmental and public health studies. The manuscript reports on observational records in Barcelona, Spain.

Thank you very much for the kind comments to our work.

General comments.

Recommendations for practice and policy: A potential key contribution of this study could be about offering recommendations for practice and policymaking; for example, calling for informing stakeholders, including public and healthcare managers, through media and education regarding the effects of SHS exposure and/or the need to consider proximity distance limitations to create a fully smoke-free zone around schools and healthcare environments etc.

Thank you very much for the comments. We agree with them. As the reviewer suggests, we have modified the discussion section as follows:

“One third of terraces with children present had smokers seated at the same table as the children, showing a disregard for their own children’s health by the caretakers. Furthermore, children do not have control over their environment or consciousness of dangers, meaning they need protection by a third party, caretakers or the legislation [4]. These kinds of legislations should be implemented in Spain to protect vulnerable populations (i.e. children, pregnant women, elderly), reinforcing the idea of promoting fully smoke-free zones around schools and healthcare environments. The smoking prohibition should be equated for the e-cigarette in all smoke-free zones as well [18]. Awareness campaigns would also help make the population gain consciousness about the relevance of the issue.”

Suggestion for the authors: It would be helpful to include one of the completed ‘tobacco worksheets’ (line 83) as an example of observation carried out with this method (perhaps in the appendix). This helps the reader understand the method and details of data collection.

Thank you very much for the suggestion. We have modified it* and included a second appendix with the data collection sheet, as follows:

Appendix B

Figure 2. Tobacco worksheet for data collection.

* Image enlarged at the end of the document.

Minor issues:

Consistency in the citation style

Page 1, line 35: 1 should read [1]

Page 1, line 44: 9 should read [9]

Page 2, line 47: 9 should read [9]

Thank you very much for the comment. The text has been carefully reviewed and corrected, updating both the references and the citation style in the text. It was a mistake when transferring the references from the original Word to the Healthcare template that has now been corrected.

Round 2

Reviewer 1 Report

After the revision, the paper is clearer and the design is reasonable. It is suggested to modify the language properly.